# Glucokinase Variant Proteins Are Resistant to Fasting-Induced Uridine Diphosphate Glucose-Dependent Degradation in Maturity-Onset Diabetes of the Young Type 2 Patients

**DOI:** 10.3390/ijms242115842

**Published:** 2023-10-31

**Authors:** Jaeyong Cho, Yukio Horikawa, Yuki Oiwa, Kazuyoshi Hosomichi, Daisuke Yabe, Takeshi Imai

**Affiliations:** 1Department of Chemical Biology, National Center for Geriatrics and Gerontology, Obu 474-8511, Japan; jaeyong420@gmail.com (J.C.); yoiwa1987@gmail.com (Y.O.); 2Departments of Diabetes, Endocrinology and Metabolism, Gifu University Graduate School of Medicine, Gifu 501-1194, Japan; horikawa.yukio.y5@f.gifu-u.ac.jp (Y.H.); yabe.daisuke.s9@f.gifu-u.ac.jp (D.Y.); 3Laboratory of Computational Genomics, School of Life Science, Tokyo University of Pharmacy and Life Sciences, Tokyo 192-0392, Japan; khosomic@toyaku.ac.jp; 4Department of Rheumatology and Clinical Immunology, Gifu University Graduate School of Medicine, Gifu 501-1194, Japan

**Keywords:** maturity-onset diabetes of the young type 2, uridine diphosphate glucose, cereblon, glucokinase

## Abstract

We previously reported that glucokinase undergoes ubiquitination and subsequent degradation, a process mediated by cereblon, particularly in the presence of uridine diphosphate glucose (UDP-glucose). In this context, we hereby present evidence showcasing the resilience of variant glucokinase proteins of maturity-onset diabetes of the young type 2 (MODY2) against degradation and, concomitantly, their influence on insulin secretion, both in cell lines and in the afflicted MODY2 patient. Hence, glucose-1-phodphate promotes UDP-glucose production by UDP-glucose pyrophosphorylase 2; consequently, UDP-glucose-dependent glucokinase degradation may occur during fasting. Next, we analyzed glucokinase variant proteins from MODY2 or persistent hyperinsulinemic hypoglycemia in infancy (PHHI). Among the eleven MODY2 glucokinase-mutated proteins tested, those with a lower glucose-binding affinity exhibited resistance to UDP-glucose-dependent degradation. Conversely, the glucokinase^A456V^-mutated protein from PHHI had a higher glucose affinity and was sensitive to UDP-glucose-dependent degradation. Furthermore, in vitro studies involving UDP-glucose-dependent glucokinase variant proteins and insulin secretion during fasting in Japanese MODY2 patients revealed a strong correlation and a higher coefficient of determination. This suggests that UDP-glucose-dependent glucokinase degradation plays a significant role in the pathogenesis of glucose-homeostasis-related hereditary diseases, such as MODY2 and PHHI.

## 1. Introduction

The principal target of glucose in glucose-induced insulin secretion (GSIS) resides in glucokinase (GCK [1]), an enzyme tasked with the phosphorylation of glucose, ultimately generating glucose-6-phosphate (G6P). As such, glucokinase assumes a pivotal role within the GSIS framework. Monogenic and dominant glucokinase variants (Online Mendelian Inheritance in Man, OMIM *138079 (accessed on 28 October 2023) have been observed in several diseases, including maturity-onset diabetes of the young type 2 (MODY2 OMIM#138079 [2,3,4,5,6,7,8,9]), permanent neonatal diabetes mellitus (PNDM, OMIM#606176), and persistent hyperinsulinemic hypoglycemia in infancy (PHHI, OMIM #256450, #601820, #602485, #609975, #609968, #606762, #610021 [10,11,12,13,14]). PHHI type 3 is associated with the glucokinase variant A456V [11]. In addition to these glucokinase variants, we have discerned instances of glucokinase copy number aberrations within the realm of oncology. Notably, glucokinase gene amplification has surfaced in approximately one percent of pan-cancer cases (1.230% = 203 cases/16382 total cancer cases @ cBioportal, 0.758% = 83 cases/10953 total @ The Cancer Genome Atlas (TCGA)). Conversely, homozygous deep deletion of glucokinase remains a rare phenomenon (0 cases/16382 total @ cBioportal, 7 cases/10953 total @ TCGA). This gain-of-function alteration of glucokinase in cancer pathogenesis is theorized to augment glycolytic activity, a trait commonly exhibited by rapidly proliferating tumor cells [15]. Conversely, the suppression of glycolysis within neoplastic cells is commonly referred to as the Warburg effect. Collectively, it is apparent that glucokinase gene mutations do not merely confound diabetes mellitus but also extend their influence over a spectrum of other pathological conditions.

Furthermore, glucokinase is one of the substrate proteins of E3 ubiquitin ligase, cereblon in the cullin ring ubiquitin ligase 4 (CRL4) complex [1,16,17,18,19,20], and UDP-glucose is the ligand of cereblon for glucokinase ubiquitination and degradation [21,22,23,24]. UDP-glucose acts as an endogenous degron, protein degrader, proteolysis-targeting chimera (PROTAC), or molecular glue. It serves as the molecular bridge that connects cereblon to glucokinase, thereby instigating glucokinase’s ubiquitination and ultimate degradation [21]. The degron structure consists of three integral components: ubiquitin ligase binding, linker, and substrate protein binding. Prior work has been undertaken to optimize ubiquitin ligase binding and linker length [21]. The final component, glucokinase binding, comprises glucose derivatives [21], despite the fact that glucokinase’s enzymatic activity is primarily focused on glucose among various monosaccharides. It should be noted that our attempts to optimize the glucokinase-binding moiety within the degron structure have been met with challenges [21].

In this context, we have unveiled an alternative approach to enhance the optimization of mutated substrate proteins, notably exemplified by the A456V-glucokinase variant, which exhibited a heightened affinity for glucose in comparison to its wild-type counterpart. Additionally, we have determined that glucokinase proteins carrying MODY2-associated mutations typically display diminished glucose-binding affinities. Furthermore, the glucose affinity exhibited by glucokinase variant proteins is intimately linked to UDP-glucose-dependent glucokinase degradation within β-cell lines, as well as insulin secretion dynamics in both β-cell lines and MODY2 patients.

## 2. Results

### 2.1. The Correlation between Low-Glucose-Dependent Insulin Secretion in MODY2 Patients and In Vitro

To commence, we scrutinized the clinical data pertaining to thirty individuals afflicted with MODY2 (Figure 1 and Appendix A). Prior to the glucose tolerance test, we assessed fasting plasma glucose (FPG in mg/dL) and fasting immunoreactive insulin (FIRI in μg/mL). It is worth noting that glucose tolerance tests are typically eschewed in diabetic patients; thus, the measurements were obtained during the patients’ initial visit [6]. We have presented the individual values of the homeostatic model assessment of β-cells (HOMAβ, Figure 1a), the homeostatic model assessment for insulin resistance (HOMA-IR, Appendix A), and the insulinogenic index for MODY2 patients (Figure 1b). These values were arranged in ascending order of HOMAβ, revealing that the M235T variant exhibited the lowest HOMAβ values. Out of the thirty glucokinase variants, we meticulously selected eleven variants (M235T, E256K, E442*, G264S, G261R, S336F, G193F, L388P, G162R, R43C, and T206M) for in vitro analysis. The insulin secretion activity was analyzed under both low glucose conditions (Figure 1c,d) and high glucose conditions (Figure 1e). These in vitro analyses are analogous to HOMAβ during fasting and the insulinogenic index following the glucose tolerance test.

A substantial correlation between HOMAβ and low-glucose-induced insulin secretion was unveiled (Figure 1c,d). The insulinogenic index (IRI30-IRI0/FPG30-FPG0) quantifies the insulin secretion capacity following a 30 min glucose tolerance test, signifying its similarity to insulin secretion under high glucose conditions. It is imperative to highlight that the insulinogenic index of the eleven MODY2 patients was compared with insulin secretion during high glucose exposure in β-cell lines. Regrettably, a weaker correlation was detected (Figure 1e). This implies that the MODY2 glucokinase variant is not implicated in high-glucose-induced insulin secretion in β-cell lines, even though it is involved in insulin secretion under low glucose conditions in vitro (Figure 1f).

### 2.2. Upstream of UDP-Glucose Pyrophosphorylase 2 (UGP2): Low-Glucose-Induced UDP-Glucose-Dependent Glucokinase Ubiquitination and Degradation

The glucokinase variants are unquestionably implicated in insulin secretion under fasting or low glucose conditions, albeit not in the context of glucose tolerance or high glucose levels (Figure 1). Our previous research has elucidated that glucokinase mutations contribute to UDP-glucose-dependent glucokinase degradation [1,21]. Our pursuit is dedicated to unraveling the intricate relationship between fasting, low glucose, and UDP-glucose. UGP2 stands as the exclusive enzyme responsible for the synthesis of UDP-glucose from glucose-1-phosphate (G1P) and uridine triphosphate (UTP) [21]. Our previous findings have unequivocally demonstrated that UGP2 plays a pivotal role in UDP-glucose-dependent glucokinase protein degradation through a UGP2-knockdown (UGP2-KD) system [23], underscoring the consequential impact of UGP2 activity on UDP-glucose-dependent glucokinase degradation.

To delve into the biological significance of UGP2 activity, we embarked on an exploration of the upstream processes concerning UGP2 substrates, namely G1P and UTP. In response to fasting, glycogenolysis is induced [21], precipitating a reduction in glucose levels and concurrent depletion of various nutrients, including arginine [1]. Within the NIT-1 β-cell line, we meticulously conducted experiments utilizing both high glucose (24.98 mM) and low glucose (0.36 mM) treatment regimens. The intracellular concentrations of G1P (Figure 1a) and UDP-glucose (Figure 1b) were markedly elevated under low glucose conditions, while intracellular G6P concentration (Figure 1c) and secreted insulin (Figure 1d) were significantly diminished during low glucose treatment.

Subsequently, we conducted a comprehensive investigation of the UGP2-KD system ([21,23], Figure 1, and Appendix A). Administration of G1P yielded a pronounced reduction in glucokinase protein levels in the presence of UGP2 (Figure 1e, Appendix A). Correspondingly, G1P administration led to reduced intracellular G6P levels and attenuated insulin secretion in UGP2-WT cells. Conversely, G1P administration to UGP2-KD cells failed to exert any discernible or statistically significant influence on glucokinase protein levels, intracellular G6P, or insulin secretion. Furthermore, the introduction of UDP-glucose resulted in diminished insulin secretion in both UGP2 WT and KD cells [21,23]. In summation, low glucose precipitates G1P production through glycolysis, subsequently augmenting UGP2 activity and facilitating the generation of UDP-glucose. This induced UDP-glucose, in turn, instigates cereblon-dependent glucokinase protein degradation.

### 2.3. High Correlation among Glucose-Binding, UDP-Glucose-Dependent Degradation and Insulin Secretion under Low Glucose in MODY2 Glucokinase Variant Proteins In Vitro

Primarily, the enhancement of degron/PROTAC compounds revolves around optimizing substrate-protein-binding moieties, linker design, and ubiquitin ligase binding motifs [21]. Among the three moieties, our focus extended to linker length and ubiquitin ligase cereblon-binding moieties [21]. It is noteworthy that the last moiety, which binds to glucokinase, comprises glucose derivatives [21], despite the fact that glucokinase solely phosphorylates glucose among monosaccharides. It is regrettable to report that our endeavors to optimize the glucokinase-binding motif yielded no tangible progress [21]. Consequently, we ventured down an alternative path of optimization, concentrating on the glucokinase protein itself.

As elucidated earlier, glucokinase mutant proteins are encountered in MODY2 [6,7] and PHHI [12,14,15,16]. Our selection encompassed eleven glucokinase variants (M235T [3,10], E256K [25,26], E442* [27], G264S, G261R, S336F, G193F, L336P, G162F, R43C, and T206M [12,28]) from MODY2 (Figure 2). In addition, we considered the A456V variant from PHHI (Appendix A). Initially, we conducted an assessment of hyperinsulinemia associated with the PHHI A456V variant, comparing it to WT-glucokinase (Appendix A). Remarkably, the A456V [12,14,15,16] mutant protein exhibited heightened glucose affinity (Appendix A), accompanied by reduced glucokinase protein levels (Appendix A), decreased G6P production (Appendix A), and diminished insulin secretion (Appendix A) in a UDP-glucose-dependent manner. These findings underscore the heightened sensitivity of the A456V mutant protein to UDP-glucose-dependent glucokinase degradation (Appendix A). This degradation, influenced by UDP-glucose, may be a contributing factor to the pathogenesis of PHHI.

Subsequently, we embarked on an in-depth analysis of the eleven glucokinase variants derived from hypoinsulinemia MODY2 (Figure 3). We assessed their glucose-binding activities, ordering them based on their respective glucose affinities (Figure 3a). The eleven glucokinase variants were ranked according to their individual glucose-binding affinities, ranging from the lowest (M235T = 10.78%, E256K = 27.65%, E442* = 23.15%, G264S = 35.61%, G261R = 33.20%, S336F = 36.60%, G193F = 52.65%, L336P = 55.60%, G162F = 56.90%, R43C = 70.31%, and T206M = 132.85%). Notably, only the T206M variant exhibited a higher glucose affinity than WT-glucokinase (100%). Subsequently, we conducted an analysis of UDP-glucose-dependent degradation (Figure 3a,b) and insulin secretion under low glucose conditions (Figure 3c,d). The data were then compared with the order of glucose-binding activity (Figure 3a) and visualized on a scatter plot to reveal correlations (Figure 3b,d). The relationship between glucose binding (*X*-axis in Figure 3b) and UDP-glucose-dependent degradation (*Y*-axis in Figure 3b) yielded a dataset with a high degree of correlation (coefficients of determination, R^2^ = 0.863). Similarly, the correlation between UDP-glucose-dependent degradation and insulin secretion under low glucose conditions exhibited a notably high coefficient of determination (R^2^ = 0.926, Figure 3d). Conversely, there were no discernible relationships between insulin secretion under low and high glucose conditions (Figure 3e).

In summary, these findings collectively underscore that glucokinase variants associated with MODY2 manifest resistance to UDP-glucose-dependent degradation, thereby contributing to hypoinsulinemia. In contrast, the PHHI variant demonstrates heightened sensitivity to UDP-glucose-dependent degradation, resulting in hyperinsulinemia (Appendix A). Notably, glucose-binding activity concurs with UDP-glucose-dependent glucokinase degradation and insulin secretion under low glucose conditions in vitro (Figure 3f). 

### 2.4. UDP-Glucose-Dependent MODY2 Glucokinase Variant Protein Degradation Was Involved in HOMAβ

Our comprehensive investigation into MODY2, spanning in vitro studies conducted in patients (Figure 1) and β-cells (Figure 2 and Figure 3), has yielded invaluable insights. In this section, we synthesize these findings to explore the relationship between UDP-glucose-dependent mutated glucokinase degradation in β-cells and insulin secretion in MODY2 patients. Eleven MODY2-mutated glucokinase variants have consistently displayed resistance to UDP-glucose-dependent degradation, culminating in reduced insulin secretion under low glucose conditions in vitro (Figure 2 and Figure 3). Furthermore, thirty MODY2 patients exhibited diminished HOMAβ values (Figure 1). We juxtaposed individual glucokinase variants (Figure 1 and Figure 3) to facilitate a direct comparison. Remarkably, the M235T variant emerged with the lowest values, both in vitro and in patients, while T206M exhibited the highest values, in line with the in vitro and patient data.

Within this context, we embarked on a comparative analysis between in vitro data and clinical data for individual MODY2 variants (Figure 4). The eleven variants (M235T, E256K, E442*, G264S, G261R, S336W, G193W, L386P, G162R, R43C, and T205M) from Figure 1 were juxtaposed with Japanese MODY2 variants (Figure 3). Glucose-binding affinity was ranked in ascending order based on individual values (Figure 4a,b). Notably, UDP-glucose-dependent degradation (in blue) and HOMAβ (indicated in orange) exhibited similar trends across individual variants (bar graph in Figure 4a and scatter plot in Figure 4b). The correlations between insulin secretion in MODY2 patients and insulin secretion in β-cells with glucokinase variant proteins were highly congruent. Hence, our in vitro findings likely mirror the dynamics occurring within the pancreas of MODY2 patients. It is plausible that UDP-glucose-dependent glucokinase degradation occurs during fasting in humans with WT-glucokinase, but this process appears to be impaired in glucokinase variants associated with MODY2 and PHHI.

## 3. Discussion

### 3.1. Glucose-Binding and UDP-Glucose-Dependent Degradation

UDP-glucose and glucokinase stand as the primal examples of endogenous cereblon ligands and their corresponding substrate proteins [21]. Uridine engages cereblon akin to thalidomide, employing diphosphate as a bridging agent, while glucose assumes the mantle of the moiety tasked with glucokinase binding [21]. UDP-glucose operates as a molecular adhesive, facilitating the interaction between glucokinase and cereblon. In our investigation involving twelve mutated glucokinase proteins, we observed noteworthy coefficients of determination between glucose-binding affinity and UDP-glucose-dependent degradation (Figure 3 and Appendix A). These findings underscore a robust correlation between glucokinase binding and its subsequent degradation.

### 3.2. Glucose-Binding Activity of Glucokinase Variants

The glucose-binding activity exhibited variations among glucokinase variants originating from MODY2 and PHHI (Figure 1 and Figure 3 and Appendix A). With the exception of T206M, ten glucokinase variants displayed diminished glucose-binding activity compared to the wild-type (WT) form. Notably, the A456V variant from PHHI and the T206M variant demonstrated pronounced affinities for glucose (Figure 3 and Appendix A). It is worth noting that PHHI is associated with hyperinsulinemia [11,12,13,14], and T206M exhibited the highest HOMAβ and fasting immunoreactive insulin (FIRI) values among thirty Japanese MODY2 patients (Figure 1 and Appendix A). Intriguingly, the in vitro glucose-binding activity exhibited a positive correlation with insulin secretion in patients. This correlation might be attributed to compromised UDP-glucose-dependent glucokinase degradation in MODY2 or PHHI patients.

### 3.3. UDP-Glucose as a Therapeutic Drug

Devising innovative avenues for drug discovery, degrons/PROTACs have emerged as a promising frontier [24]. In this context, we posit the therapeutic potential of UDP-glucose for treating PHHI. Given the robust binding of the A456V variant protein to glucose (Appendix A) and the fact that PHHI patients possess one allele of glucokinase^WT^ and one allele of glucokinase^A456V^, UDP-glucose exhibits a predilection for degrading A456V over the WT (Appendix A). Administration of UDP-glucose to PHHI patients resulted in a higher prevalence of WT-glucokinase relative to A456V-glucokinase (Appendix A), consequently favoring the WT variant. UDP-glucose holds promise as a therapeutic avenue for mitigating hyperinsulinemia in PHHI.

## 4. Materials and Methods

### 4.1. Reagents

Glucose-1-phosphate (G1P, QB-6893, Wako Pure Chemicals Industries, Ltd., Osaka, Japan) and UDP-glucose (Wako Pure Chemicals Industries, Ltd., Osaka, Japan) were procured [21]. These subsequent antibodies were acquired: β-actin (sc-47778, Santa Cruz Biotechnology, Santa Cruz, CA, USA), FLAG (F-1804, Sigma, St. Louis, MO, USA), and UDP-glucose pyrophosphorylase 2 (UGP2, sc-377089, Santa Cruz Biotechnology, Santa Cruz, CA, USA). UGP2 knockdown siRNA (sc-154894) was procured from Santa Cruz Biotechnology (Santa Cruz, CA, USA).

### 4.2. Cell Culture

Mouse-pancreas-derived NIT-1 cells (CRL-2055TM obtained from ATCC, Manassas, VA, USA) were cultured as previously described [1,21,23]. Succinctly, cells were seeded at a density of 1.5–3.0 × 10^6^ cells/60 mm dish with substitution of F-12K medium (Kaighn’s modification of Ham’s F-12 medium) supplemented with 10% fetal calf serum (FCS) after 48 h of incubation. Human embryo kidney 293T (HEK293T) cells and human hepatocellular carcinoma (Hep G2) cells were sustained with DMEM + 10% FCS. Cereblon KO cells were generously provided by Tokyo Medical University [21,23,29]. Glucose concentrations in various media were as follows: DMEM (0.5551 mM, 100 mg/dL), high-glucose DMEM (24.978 mM, 450 mg/dL), F12K (0.699 mM, 126 mg/dL), and low glucose (0.3608 mM, 65 mg/dL), respectively.

### 4.3. Analysis of Insulin Secretion from Cells

Insulin secretion was ascertained using a commercial enzyme-linked immunosorbent assay (ELISA) kit (Shibayagi, Gunma, Japan) as previously detailed [1,21,23]. In brief, the culture medium of the NIT-1 cells was substituted with UDP-glucose-containing F12K + 10% fetal calf serum (FCS) for the indicated duration. Subsequently, the medium was harvested for insulin quantification. 

### 4.4. Chemical Detection

Intracellular G6P and UDP-glucose concentrations were determined through two methodologies [1]. One method involved the utilization of a commercial kit (Glucose-6-Phosphate Fluorometric Assay Kit No. 700750, Cayman chemical, Ann Arbor, MI, USA [1]). The other approach encompassed metabolome analysis [1]. Chemicals were analyzed using the following commercial kits: glucose; 346-09411 Wako, G1P; Sigma MAK098-1KT, G6P; glucose-6-phosphate fluorometric assay kit, No. 700750, Cayman chemical, Ann Arbor, MI, USA. Other parameters were assessed through metabolome analysis [21].

### 4.5. Immunoprecipitation and Western Blot

Immunoprecipitation and Western blot analyses were executed as previously elucidated, with minor adaptations [1,21,23]. NIT-1 cells transfected with the pCDNA3-insulin-Myc expression vector were cultured in arginine-free medium for 30 min prior to arginine supplementation. Cells were lysed in a buffer comprising 20 mM HEPES-NaOH (pH 7.9), 1 mM MgCl_2_, 0.2 mM CaCl_2_, 100 mM KCl, 0.2 mM EDTA, 10% glycerol, 0.1% Nonidet P-40, 1 mM dithiothreitol, 0.2 mM phenylmethylsulfonyl fluoride (Nacalai Tesque, Kyoto, Japan), and 3% n-octyl-β-D-glucoside (DOJINDO, Tokyo, Japan). After incubation on ice for 30 min, lysates were centrifuged at 12,000× *g* for 15 min at 4 °C and dialyzed for 3 h at 4 °C in a lysis buffer devoid of n-octyl-β-D-glucoside. Subsequent to dialysis, the supernatant was incubated with the specified antibody for 120 min at 4 °C. The samples were subsequently incubated with Protein G Sepharose 4 Fast Flow (GE Healthcare, Chicago, IL, USA). After additional washing, the precipitates were heat-denatured in SDS-sample buffer. For immunoblot analysis, proteins were segregated by SDS-PAGE and transferred onto a PVDF membrane for Western blotting. The membranes were obstructed with 5% non-fat milk and Tris-buffered saline with Tween 0.1%, incubated overnight with a mixture of primary antibody (c-Myc (1:1000)) and Can Get Signal solution (TOYOBO, Osaka, Japan) at 4 °C, washed, incubated with the secondary antibody for 60 min at room temperature, and subsequently washed again. Immune complexes were detected utilizing Immunostar LD (Wako Pure Chemicals Industries, Ltd., Osaka, Japan) substrate. Signals were quantified using the LAS 4000 imaging system (GE Healthcare, Chicago, IL, USA). Image J (Wayne Ras-band (NIH), WC, USA) was employed for the densitometric analysis of scanned membranes [1,21,23].

### 4.6. Immunocytochemical Analysis

Human embryo kidney 293T (HEK293T) cells were nurtured in Dulbecco’s modified Eagle’s medium (Gibco-Thermo Fisher Scientific, Waltham, MA, USA) supplemented with 10% heat-inactivated fetal bovine serum, 1% non-essential amino acids solution (NEAA, 139-15651, Wako Pure Chemicals Industries, Ltd., Osaka, Japan), and 1% antibiotic–antimycotic mixed stock solution at 37 °C and 5% CO_2_. For transfection, HEK293T cells were seeded into 12-well plates with glass coverslips coated with 0.0004% poly-*L*-lysine (Sigma, St. Louis, MO, USA) at a density of 3 × 10^5^ cells per well one day prior to transfection. pCDNA3-GCKHA vectors were transfected into HEK293T cells using Lipofectamine (Thermo Fisher Scientific, Waltham, MA, USA) according to the manufacturer’s protocol. Four hours after transfection, the medium was replaced with fresh medium based on the experimental conditions. If necessary, further medium changes were performed depending on the experiment.

Transfected cells were washed twice with PBS (phosphate-buffered saline) and subsequently fixed with 4% paraformaldehyde for 10 min at room temperature. Cells were washed thrice with PBS and then permeabilized in 0.1% Triton-X 100 diluted in PBS for 10 min at room temperature. After permeabilization, cells were washed thrice with PBS for five minutes each before blocking with PBST (PBS + 0.1% Tween 20) containing 5% bovine serum albumin (BSA) for one hour at room temperature. The cells were incubated overnight at 4 °C with anti-HA high affinity (3F10) (Roche, Basel, Switzerland) diluted 1:1500 in PBST containing 1% BSA. Subsequently, the cells were washed thrice with PBS, followed by incubation with F(ab’)2-Goat anti-Mouse IgG (H + L) Cross-Adsorbed Secondary Antibody, Alexa Fluor™ 555 (Thermo Fisher Scientific, Waltham, MA, USA) and Hoechst 33,258 (Dojindo, Kumamoto, Japan), both diluted 1:1000 in PBST with 1% BSA, for one hour at room temperature. Cells were washed thrice with PBS for 5 min each before mounting with VECTASHIELD mounting medium (Vector Laboratories, CA, USA). Images were acquired using a BZ-X 800 fluorescence microscope (KEYENCE, Osaka, Japan). Analysis of the acquired images was performed using a BZ-X 800 analyzer.

### 4.7. MODY Study

A total of 230 patients with diabetes were recruited from 75 medical institutions throughout Japan, whose disease was suspected to be MODY. Of these, 103 were male and 127 were female, with a mean age of onset of 16.2 (SD 7.0) years and a mean BMI of 21.8 (SD 3.8) kg/m^2^. The inclusion criteria comprised an onset age of ≤35 years, autoimmune antibody negativity, and non-obesity (BMI < 30 kg/m^2^). Family history was not considered to avoid overlooking sporadic or low-penetrant cases. The study protocol received approval from the institutional review board of Gifu University (No. 25–153), and informed consent was obtained from all participants. Family history was excluded to ensure the inclusion of sporadic or low-penetrant cases [6].

### 4.8. Ethical Approval

The test protocol was sanctioned by the Institutional Review Board of Gifu University (Gifu-No. 25–153) in accordance with the Declaration of Helsinki. Written informed consent was obtained from all participants [6]. 

### 4.9. Statistical Analysis

In both in vitro analyses and the clinical study, we employed one-way analysis of variance (one-way ANOVA) and the single-sided Student’s *t* test. The values are presented as means ± standard error (*n* = 6). Statistical significance (single-sided Student’s *t* test) is denoted in the figure legends as follows: * *p* < 0.05. *n* = 6 for statistical analysis [1,21,23]. For the reproducibility of key experiments, we conducted a total number of experiments exceeding five times, including experiments for establishing conditions with comparable results. Furthermore, we employed multiple approaches to corroborate a singular outcome.

## 5. Conclusions

Our prior study illuminated the ubiquitination and degradation of glucokinase, orchestrated by cereblon with the indispensable aid of UDP-glucose [21]. UDP-glucose assumes the role of a molecular adhesive, acting as the linchpin in mediating the interplay between cereblon and glucokinase. Glucokinase variants characterized by a diminished affinity for glucose manifest resistance to UDP-glucose-dependent degradation, both in cell lines and among patients. Conversely, variants displaying heightened glucose affinity exhibit a susceptibility to UDP-glucose-dependent degradation. The process of UDP-glucose-dependent glucokinase degradation emerges as a pivotal mechanism in human physiology, and its dysfunction in MODY2 culminates in hypoinsulinemia.

## Figures and Tables

**Figure 1 ijms-24-15842-f001:**
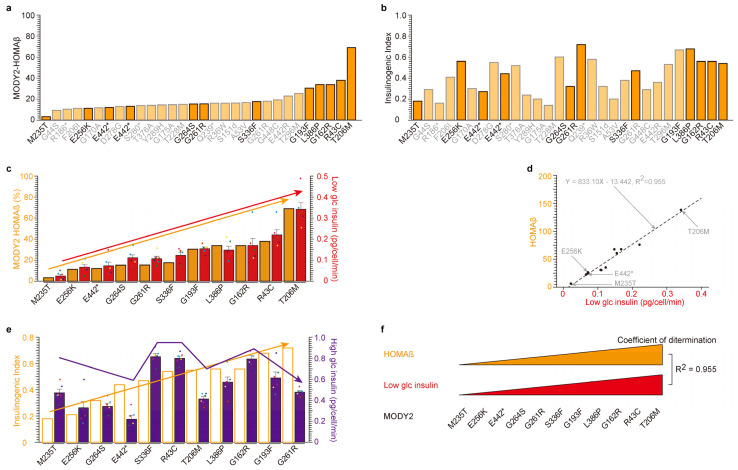
Correlation between low-glucose-dependent insulin secretion in MODY2 patients (the homeostatic model assessment of β-cells (HOMAβ)) and in vitro. (**a**,**b**) Individual values of the homeostatic model assessment of β-cells (HOMAβ, (**a**)) and insulinogenic index (**b**) of thirty glucokinase variant proteins from Japanese MODY2 patients were presented in ascending order based on their insulin secretion activity during fasting (**a**). Eleven glucokinase-mutated proteins in strong orange bars (M235T, E256K, E442*, G264S, G261R, S336F, G193F, L388P, G162R, R43C, and T206M) were analyzed in vitro. *: stop codon (**c**,**d**) Similar tendency of insulin secretion under low glucose in MODY2 patients (orange bars) and in vitro analysis (red bars) among eleven mutants. The eleven glucokinase variants were in ascending order of HOMAβ activity (left orange *Y*-axis, (**c**)). The values are reported as the means ± standard error (*n* = 6) with six individual data point. Coefficient of determination; R^2^ = 0.955. (**e**) No correlation of insulin secretion under high glucose in MODY2 patients (empty bars) and in vitro analysis (purple bars) among eleven mutants. The eleven glucokinase variants were in ascending order of insulinogenic index (left orange *Y*-axis). The values are reported as the means ± standard error (*n* = 6) with six individual data point. (**f**) Graphic summary of Figure 1. There are highly significant correlations between insulin secretion under low glucose conditions in patients and in vitro that are dependent on glucokinase variants.

**Figure 2 ijms-24-15842-f002:**
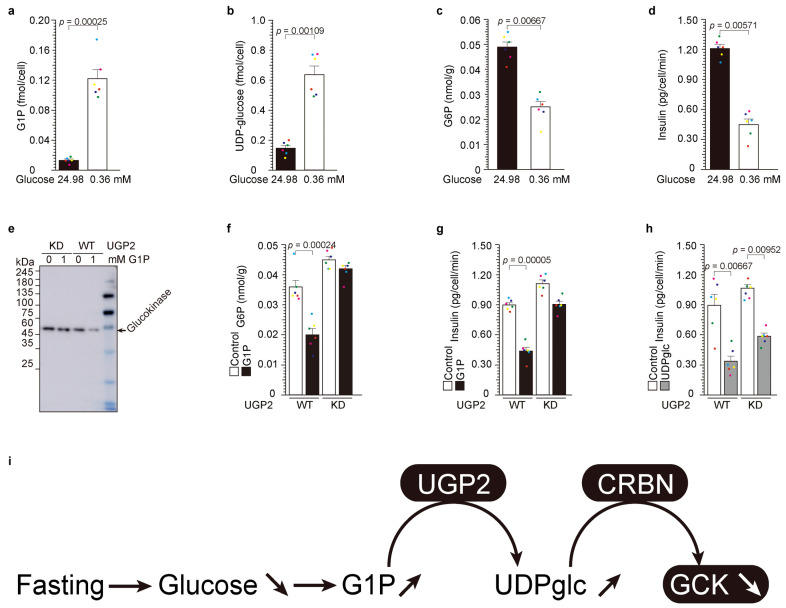
Low-glucose-induced UDP-glucose-dependent glucokinase ubiquitination and degradation. (**a**–**d**) NIT-1 cells were cultivated in media containing 24.98 mM (450 mg/dL) and 0.36 mM (65 mg/mL) glucose for a duration of 3 h. The values are reported as the means ± standard error (*n* = 6) with six individual data point. (**e**–**g**) The administration of G1P had no discernible effect on the quantity of glucokinase protein (**e**), G6P levels (**f**), or the secretion of insulin (**g**) in UGP2KD cells [1]. (**h**) The introduction of UDP-glucose resulted in a reduction in insulin secretion in both UGP2-WT and KD cells ([25] and (**h**)). The values are reported as the means ± standard error (*n* = 6) with six individual data point. (**i**) Graphic summary of Figure 2.

**Figure 3 ijms-24-15842-f003:**
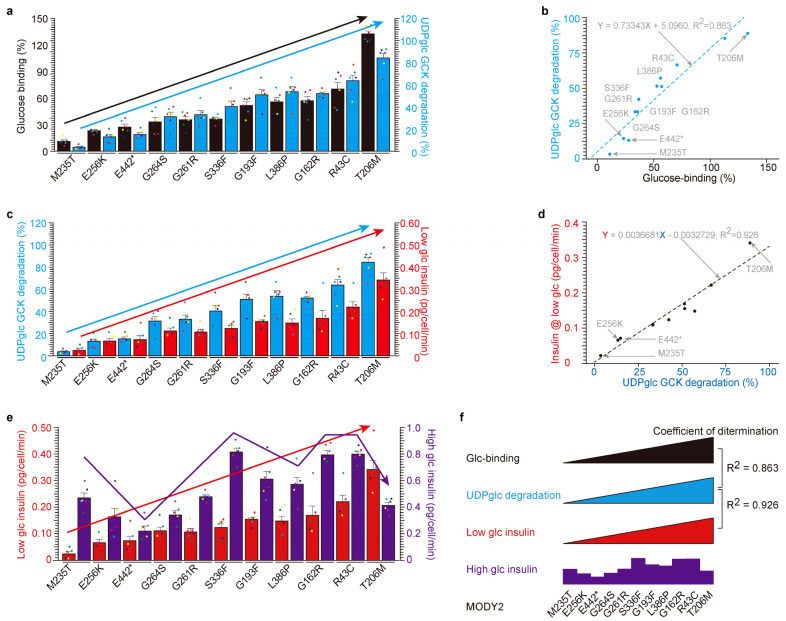
High correlation among glucose-binding, UDP-glucose-dependent degradation and insulin secretion under low glucose in MODY2 glucokinase variant proteins in vitro. (**a**,**b**) The glucose-binding activities of eleven MODY2 glucokinase variants (filled black bar) were correlated with their susceptibility to UDP-glucose-induced degradation (blue bar). The eleven glucokinase variants are in ascending order of glucose-binding activity (left black *Y*-axis, (**a**)). The values are reported as the means ± standard error (*n* = 6) with six individual data point. UDP-glucose-dependent glucokinase degradation (right blue *Y*-axis, (**a**)). The scatter plot between glucose-binding % (*X*-axis) and UDP-glucose-dependent degradation (*Y*-axis) yielded a highly correlated dataset (coefficients of determination, R^2^ = 0.863). (**c**,**d**) The activities of UDP-glucose-dependent degradation of eleven MODY2 glucokinase variants (filled blue bars) were correlated with their susceptibility to insulin secretion under low glucose conditions (red bars). The eleven glucokinase variants are in ascending order of UDP-glucose-dependent degradation activity (same as glucose-binding activity, left blue *Y*-axis, (**c**)) and low glucose insulin secretion (right red *Y*-axis, (**c**)). The values are reported as the means ± standard error (*n* = 6) with six individual data point. The scatter plot between UDP-glucose-dependent degradation (*X*-axis) and low glucose insulin secretion (*Y*-axis) yielded a highly correlated dataset (coefficients of determination, R^2^ = 0.925). (**e**) The low glucose insulin secretion of eleven MODY2 glucokinase variants (filled red bar) was not correlated with their susceptibility to insulin secretion with high glucose (purple bar). Eleven glucokinase variants are in ascending order of low glucose insulin secretion activity (left red *Y*-axis) and high glucose insulin secretion (right purple *Y*-axis). The values are reported as the means ± standard error (*n* = 6) with six individual data point. (**f**) Summary of Figure 2. There were high correlations between glucose-binding, UDP-glucose-dependent degradation and insulin with low glucose, but no correlation between low- and high-glucose-induced insulin secretion depends on glucokinase variants. *: stop codon.

**Figure 4 ijms-24-15842-f004:**
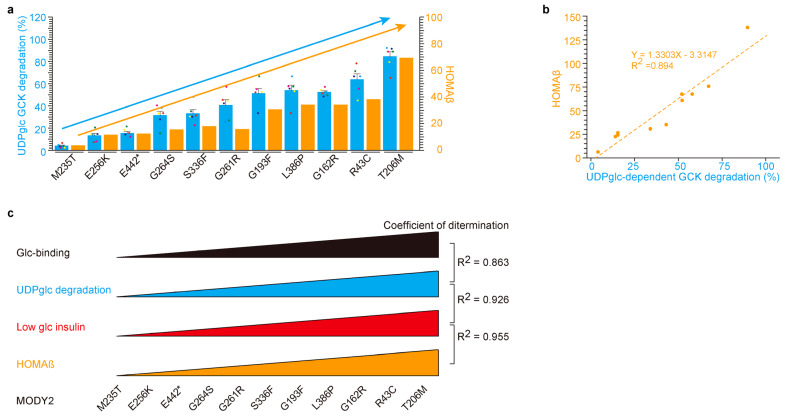
UDP-glucose-dependent MODY2 glucokinase variant protein degradation was involved in HOMAβ. (**a**,**b**) A similar relation between UDP-glucose degradation in vitro and HOMAβ in patients by glucokinase variants. Eleven glucokinase variants are in ascending fashion based on UDP-glucose-dependent degradation activity in vitro (left blue *Y*-axis, (**a**)). HOMAβ in the MODY2 patients (right orange *Y*-axis, (**a**)). The values are reported as the means ± standard error (*n* = 6) with six individual data point. The scatter plot between insulin with UDP-glucose-dependent degradation in vitro (*X*-axis, (**b**)) and HOMAβ in patients (*Y*-axis, (**b**)) yielded a highly correlated dataset (coefficients of determination, R^2^ = 0.894). (**c**) Graphic summary of Figure 4. The eleven glucokinase-mutated proteins in MODY2 exhibited resistance to UDP-glucose-dependent degradation and were associated with HOMAβ.

## Data Availability

Requests for data and materials should be addressed to T.I. (timai@ncgg.go.jp).

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
