# Peer review of "Glucokinase Variant Proteins Are Resistant to Fasting-Induced Uridine Diphosphate Glucose-Dependent Degradation in Maturity-Onset Diabetes of the Young Type 2 Patients"

_ijms, 2023, doi:10.3390/ijms242115842_

Round 1

Reviewer 1 Report

Comments and Suggestions for Authors

The submitted manuscript describes a series of experiments suggesting that UDP-glucose-dependent glucokinase degradation plays a significant role in MODY2 and PHHI. Unfortunately, the results are too poorly presented for the reader to ascertain whether the conclusions drawn are valid. The legends of the figures inadequately describe the graphs presented making them very difficult to examine. The text also offers no description of the data presented, tending more towards a discussion. The statistical comparisons are inadequately described for all data sets, and perusing the figures compounds the confusion, making it hard to tell if the correct analyses have been applied. Moreover, P values have only been presented for Figure 1. The N numbers for the experiments have not been detailed, and if the dots presented in the figures are individual data points then the N values are far too small for the experiments. This needs improving. The conclusions about MODY 2 patients and PHHI are misleading. MODY 1, 2 and 3 seem to have been studied, with no justification presented, and PHHI not at all. The methods and results need a complete rewrite to clarify the presentation of the data, which likely requires further work to increase the N number and a review of the statistical analyses. 

Author Response

Reviewer 1

Open Review

(x) I would not like to sign my review report

( ) I would like to sign my review report

Quality of English Language

( ) I am not qualified to assess the quality of English in this paper

( ) English very difficult to understand/incomprehensible

( ) Extensive editing of English language required

( ) Moderate editing of English language required

( ) Minor editing of English language required

(x) English language fine. No issues detected

Yes        Can be improved Must be improved             Not applicable

Does the introduction provide sufficient background and include all relevant references?

( )          ( )          (x)         ( )

Are all the cited references relevant to the research?

(x)         ( )          ( )          ( )

Is the research design appropriate?

( )          ( )          (x)         ( )

Are the methods adequately described?

( )          ( )          (x)         ( )

Are the results clearly presented?

( )          ( )          (x)         ( )

Are the conclusions supported by the results?

( )          ( )          (x)         ( )

Comments and Suggestions for Authors

The submitted manuscript describes a series of experiments suggesting that UDP-glucose-dependent glucokinase degradation plays a significant role in MODY2 and PHHI. Unfortunately, the results are too poorly presented for the reader to ascertain whether the conclusions drawn are valid. The legends of the figures inadequately describe the graphs presented making them very difficult to examine. The text also offers no description of the data presented, tending more towards a discussion. The statistical comparisons are inadequately described for all data sets, and perusing the figures compounds the confusion, making it hard to tell if the correct analyses have been applied. Moreover, P values have only been presented for Figure 1. The N numbers for the experiments have not been detailed, and if the dots presented in the figures are individual data points then the N values are far too small for the experiments. This needs improving. The conclusions about MODY 2 patients and PHHI are misleading. MODY 1, 2 and 3 seem to have been studied, with no justification presented, and PHHI not at all. The methods and results need a complete rewrite to clarify the presentation of the data, which likely requires further work to increase the N number and a review of the statistical analyses.

*Thank you for your suggestions.

  1. a) Data presentation and organization.

We changed the panels for Fig 2 – 4, and added the last panels of summary in each figure. Figure legends and results were also changed. Please look at new version. Size of dots were also increased to identify.

  1. b) Statistical analysis

The p-values were unnecessary, except for Figure 1. Because, we need tendency of each value of each glucokinase variant. And, more than once of GTT to diabetes patients should be avoided. The number of clinical data is one.

For the reproducibility of key experiments, we conducted a total number of experiments exceeding five times, including experiments for establishing conditions with comparable results. Furthermore, we employed multiple approaches to corroborate a singular out-come.

Reviewer 2 Report

Comments and Suggestions for Authors

Glucokinase is a key enzyme in triggering insulin secretion.  The authors focus on the the degradation process of glucokinase which involves ubiquitination stimulated by UDP-glucose.  They tie in the biochemical process with a number of clinical conditions.  This work is definitely of interest.  The problem here is organization.  It is not clear which part of the presented work is new as opposed to review of the prior work of this group. So it is really difficult to follow what issues they have already answered and what questions they are now addressing.  Greatly interesting but really we need an organized presentation.

SPECIFIC COMMENTS

Collectively, it is apparent that gluco-53 kinase gene mutations do not merely confound diabetes mellitus but also extend their 54 influence over a spectrum of other pathological conditions

PLEASE PROVIDE A TABLE OF MUTATIONS AND PHENOTYPIC CONDITIONS

It should be noted that 66 our attempts to optimize the glucokinase-binding moiety within the degron structure 67 have been met with challenges [21]

PLEASE EXPLAIN

2. Results

YOU MEAN PRIOR PERTINENT FINDINGS?? OR IS THIS NEW?

CAN YOU SEPARATE OUT THE CLINICAL STUDIES WHICH ARE QUITE INTERESTING PLEASE.

Author Response

Reviewer 2

Open Review

( ) I would not like to sign my review report

(x) I would like to sign my review report

Quality of English Language

( ) I am not qualified to assess the quality of English in this paper

( ) English very difficult to understand/incomprehensible

( ) Extensive editing of English language required

( ) Moderate editing of English language required

( ) Minor editing of English language required

(x) English language fine. No issues detected

Yes        Can be improved Must be improved             Not applicable

Does the introduction provide sufficient background and include all relevant references?

( )          ( )          (x)         ( )

Are all the cited references relevant to the research?

(x)         ( )          ( )          ( )

Is the research design appropriate?

( )          ( )          (x)         ( )

Are the methods adequately described?

( )          ( )          (x)         ( )

Are the results clearly presented?

( )          ( )          (x)         ( )

Are the conclusions supported by the results?

( )          (x)         ( )          ( )

Comments and Suggestions for Authors

Glucokinase is a key enzyme in triggering insulin secretion. The authors focus on the degradation process of glucokinase which involves ubiquitination stimulated by UDP-glucose. They tie in the biochemical process with a number of clinical conditions. This work is definitely of interest. The problem here is organization. It is not clear which part of the presented work is new as opposed to review of the prior work of this group. So it is really difficult to follow what issues they have already answered and what questions they are now addressing. Greatly interesting but really we need an organized presentation.

*Thank you for your suggestions.

We changed the panels for Fig 2 – 4, and added the last panels of summary in each figure.

SPECIFIC COMMENTS

Collectively, it is apparent that glucokinase gene mutations do not merely confound diabetes mellitus but also extend their influence over a spectrum of other pathological conditions

*Thank you for your suggestion.

PLEASE PROVIDE A TABLE OF MUTATIONS AND PHENOTYPIC CONDITIONS

*Thank you for your suggestion. As new supplement table.

It should be noted that our attempts to optimize the glucokinase-binding moiety within the degron structure have been met with challenges [21]

*Thank you for your suggestion. We have tried several glucose-derivatives, such as, galactose, fructose, FDG, NAG etc. Unfortunately, none of them degraded glucokinase, except for one X.

PLEASE EXPLAIN

  1. Results

YOU MEAN PRIOR PERTINENT FINDINGS?? OR IS THIS NEW?

*Thank you for your suggestion.

Nine variants, except for E256K and E442*, were new in vitro data. All the MODY2 patient clinical data is new.

Increased eleven GCK variant relation of glucose-binding in vitro, UDP-glucose-degradation in vitro and low glucose-insulin secretion in vitro and in patients were highly coefficient.

CAN YOU SEPARATE OUT THE CLINICAL STUDIES WHICH ARE QUITE INTERESTING PLEASE.

*Thank you for your suggestion. The clinical data were listed on the Fig 3 and 4.

Round 2

Reviewer 2 Report

Comments and Suggestions for Authors

Here is a really interesting study on the rate of degradation of glucokinase in a number of mutational variants including MODY.   The problem I have with this paper is that the most interesting data is actually in the supplement.  They have a good number of patients with excellent clinical data.   They have glucokinase degradation rates with various mutations.   They have degradation rates per ubiquitination via UDP glucose.   What is so frustrating is that the presentation is backwards.   It would be nice to see this paper recast beginning with the clinical data, then proceeding to the mutations, then showing the actual UDP glucose stimulated degradation rates.   Fundamentally, it would be good for the authors to revise their presentation to achieve better communication of the importance of their results.

Author Response

Comments and Suggestions for Authors

Here is a really interesting study on the rate of degradation of glucokinase in a number of mutational variants including MODY.   The problem I have with this paper is that the most interesting data is actually in the supplement.  They have a good number of patients with excellent clinical data.   They have glucokinase degradation rates with various mutations.   They have degradation rates per ubiquitination via UDP glucose.   What is so frustrating is that the presentation is backwards.   It would be nice to see this paper recast beginning with the clinical data, then proceeding to the mutations, then showing the actual UDP glucose stimulated degradation rates.   Fundamentally, it would be good for the authors to revise their presentation to achieve better communication of the importance of their results.

* Thank you for your good suggestion. We changed the order of figures as reviewer recommended.

Fig 1 and S1: Clinical data of all MODY2 patients

Fig 2 and S2: Low glc induced G1P. UGP2 converted G1P to UDPglc.

Fig 3 and S3: Glc-binding=UDPg degradation= insulin secretion in low glc in vitro

Fig 4 : UDPg degradation = HOMAß

Round 3

Reviewer 2 Report

Comments and Suggestions for Authors

The authors now have made a persuasive case for the degradation of glucokinase to be a signficant factor in glucokinase associated maturity onset diabetes of the young.